# Glutathione Fluorescence Sensing Based on a Co-Doped Carbon Dot/Manganese Dioxide Nanocoral Composite

**DOI:** 10.3390/ma15238677

**Published:** 2022-12-05

**Authors:** Thi-Hoa Le, Hyun-Jong Lee, Quang-Nhat Tran

**Affiliations:** Department of Chemical and Biological Engineering, Gachon University, Seongnam 13120, Republic of Korea

**Keywords:** Glutathione, NPCD-MnO_2_ nanocoral composite, fluorescence resonance energy transfer (FRET), “turn on” fluorescence

## Abstract

Glutathione (GSH) is an antioxidant thiol that has a vital role in the pathogenesis of various human diseases such as cardiovascular disease and cancer. Hence, it is necessary to study effective methods of GSH evaluation. In our work, an effective GSH sensor based on a nitrogen and phosphorus co-doped carbon dot (NPCD)-MnO_2_ nanocoral composite was fabricated. In addition to utilizing the strong fluorescence of the NPCDs, we utilized the reductant ability of the NPCDs themselves to form MnO_2_ and then the NPCD-MnO_2_ nanocoral composite from MnO_4_^-^. The characteristics of the nanocoral composite were analyzed using various electron microscopy techniques and spectroscopic techniques. The overlap between the absorption spectrum of MnO_2_ and the fluorescence emission spectrum of the NPCDs led to effective fluorescence resonance energy transfer (FRET) in the nanocoral composite, causing a decrease in the fluorescent intensity of the NPCDs. A linear recovery of the fluorescent intensity of the NPCDs was observed with the GSH level raising from 20 to 250 µM. Moreover, our GSH sensor showed high specificity and sensing potential in real samples with acceptable results.

## 1. Introduction

Glutathione (GSH) is a tripeptide of glutamic acid, glycine, and cysteine. GSH is an important antioxidant produced by the liver [1]. GSH is the predominant low molecular-weight thiol in human cells with a concentration from 0.5–10 mmol/L. Most of the cellular GSH is present in the cytosol (85–90%) and the remainder exists in many organelles. The extracellular concentrations of GSH are relatively low (for example: 2–20 μmol/L in plasma) [2]. GSH combats free radicals that can damage the cells and also has a significant role in many processes in the human body, such as immune system response regulation, cell propagation control, cysteine transport and storage, and tissue building [3,4]. GSH deficiency is associated with oxidative stress, which is a main cause of aging. In addition, low levels of GSH have been reported in patient samples suffering from Alzheimer’s disease, liver damage, neurotoxicity, and cancer; hence, GSH can be considered as an important universal biomarker in the diagnosis and therapy of a range of diseases [5,6].

Many GSH sensing approaches have previously been developed and presented, such as electrochemistry [7], electrochemiluminescence [8], Raman spectroscopy [9], fluorescence [10], high-performance liquid chromatography (HPLC) [11], and colorimetry [12]. Among the mentioned methods, fluorescence-based sensing is one of the most promising methods owing to its ease of sample preparation and operational simplicity.

The fabrication of fluorescent sensors relies on intramolecular processes such as photoinduced electron transfer (PET), and photoinduced charge transfer (PCT) or intermolecular processes, including the fluorescence resonance energy transfer (FRET), and inner filter effect (IFE); these intermolecular processes relate to energy transfer between at least two independent molecules [13,14]. In this study, we fabricated a GSH fluorescent sensor using a nitrogen and phosphorus co-doped carbon dot (NPCD)-MnO_2_ nanocoral composite. The FRET between NPCDs and MnO_2_ components in the composite is the basis for the fabrication of a GSH “turn on” sensor.

Carbon dots (CDs) are well known fluorescent materials that have attracted considerable attention from researchers owing to their outstanding photoluminescence stability, high water solubility, non-toxicity, tunable surface functionalities, and favorable biocompatibility [15,16]. For heteroatom-doped carbon dots, such as NPCDs, in addition to their excellent properties, the fluorescence performance is significantly enhanced, as demonstrated in our previous study [17]. Hence, NPCDs are promising candidates for the fabrication of fluorescence sensors. There is an abundance of carboxyl (‒COOH) and hydroxyl (‒OH) groups on the surface of CDs produced by a hydrothermal synthesis [18]. The ‒OH allows the CDs to play the role of green reductants, while the negatively charged ‒COOH on the CDs can stabilize metal particles in their formation process [19,20]. Therefore, in this study, we not only focused on the fluorescent properties of NPCDs but also took full advantage of the NPCDs as potential reducing and stabilizing agents. We used NPCDs as reductants to form MnO_2_ nanocorals through a redox reaction between the NPCDs and KMnO_4_, which subsequently contributed to the formation of the NPCD-MnO_2_ nanocoral composite. MnO_2_ is known as a good quencher of fluorophores [21]; therefore, the fluorescent intensity of the composite is considerably quenched compared to that of pristine NPCDs. When GSH is added to the material solution, fluorescence recovery of the NPCDs occurs. In summary, NPCDs play two vital functions: (1) as a fluorescent source, and (2) as a reducing and stabilizing agent for MnO_2_ in the fabrication of an NPCD-MnO_2_ based FRET sensor for GSH. The GSH sensing is described in Figure 1.

## 2. Methods

### 2.1. Chemicals

MnCl_2_, KMnO_4_, (NH_4_)_2_HPO_4_, Ca(NO_3_)_2_, K_2_SO_4_, NaCl, citric acid monohydrate, glutathione, glucose, saccharose, glycine, methionine, lysine, cysteine, tryptophan, methionine, deionized water, and human serum, were obtained from Sigma-Aldrich, St. Louis, MO, USA.

### 2.2. Instruments

UV–Vis and photoluminescence (PL) spectra were obtained using a G1103A UV–Vis spectrophotometer (Agilent, Santa Clara, CA, USA) and a QuantaMaster TM 50 PTI spectrofluorometer (Photon Technology International, Birmingham, NJ, USA), respectively. X-ray photoelectron spectroscopy (XPS) was performed using an X-ray photoelectron spectrometer (PHI 5000, Kanagawa, Japan). Raman spectroscopy was performed using a Raman spectrometer (ANDOR Monora500i, Belfast, UK) with a laser excitation of 633 nm. The morphological structure and size of the samples were characterized using scanning and transmission electron microscopy (SEM; S-4700, Hitachi Ltd., Tokyo, Japan), and (TEM; Tecnai, F30S-Twin, Hillsboro, OR, USA).

### 2.3. Preparation of NPCDs

The synthesis of NPCDs was reported in our previous study [17]. A mixture including citric acid monohydrate (1 g), (NH_4_)_2_HPO_4_ (2.5 g), and deionized water (15 mL) was poured into a 30 mL Teflon-lined stainless-steel autoclave and heated at 180 °C for 4 h. After cooling to room temperature, the solution was purified using a polyethersulfone membrane (0.22 µm) and then dialyzed in a dialysis bag (MWCO: 1000 Da) for 48 h. The resulting solution was lyophilized to obtain a powdery substance.

### 2.4. NPCD-MnO_2_ Nanocoral Composite Synthesis

The NPCD solution (0.075 g/mL, 5 mL) was added dropwise to DI water (95 mL) while stirring and heating. After the mixture was heated to boiling, a MnCl_2_ solution (0.5 mL, 0.1 M) was dropped into the mixture. After 5 min, a KMnO_4_ solution (0.35 mL, 0.1 M) was continually dropped to the mixture. The solution color turned brown, indicating the formation of MnO_2_. The heating process was then stopped, and the solution was continually vibrated for 2 h, then centrifuged and washed with deionized water to eliminate impurities. The precipitate was dried at 60 °C, re-dispersed in DI water, and placed in a refrigerator at 4 °C.

### 2.5. GSH Detection

The GSH sensing experiment was performed at room temperature. First, the NPCD-MnO_2_ solution (500 µL, 0.019 g/mL) and deionized water (2500 µL) were mixed under sonication to obtain a solution. Then, a variety amount of GSH solution were added to obtain levels of 0.1–250 µM. The solutions were diluted by deionized water to reach the final volume of 3500 µL. The solutions allowed to equilibrate for 30 min before a fluorescent measurement was carried out under an excitation wavelength of 360 nm. The experiment was repeated three times at each concentration.

### 2.6. GSH Detection in Human Serum

The NPCD-MnO_2_ solution (500 µL) was dropped into deionized water, followed by human serum (30 µL). Deionized water was added to dilute the solution to 3000 µL. An amount of GSH solution was added, then diluted before fluorescent analysis. GSH sensing in the human serum samples was performed as presented in Section 2.5.

### 2.7. Selectivity

The fluorescent response of other substances was explored to determine their interference to GSH sensing and relied on NPCD-MnO_2_ composite material.

## 3. Results and Discussion

### 3.1. NPCD-MnO_2_ Nanocoral Composite Characteristics

The optical properties of the NPCD-MnO_2_ nanocoral composite were characterized using UV–Vis and photoluminescence spectroscopy. The absorption spectra of the NPCD, MnO_2_, and NPCD-MnO_2_ nanocoral composite solutions are shown in Figure 1. The NPCD solution shows two peaks at 234 and 334 nm. Pure MnO_2_, which was synthesized through the pyrolysis of KMnO_4_, shows a peak at 385 nm. The specific absorption peaks of both pure NPCDs and pure MnO_2_ appear in the spectrum of the NPCD-MnO_2_ nanocoral composite, confirming the reduction of NPCDs to form MnO_2_.

The fluorescent emission spectrum of the NPCD-MnO_2_ nanocoral composite indicates a peak placed in 450 nm corresponding to an excitation peak at 372 nm (Figure 2). Hence, there is no marked shift in the emission and excitation spectra of the NPCD-MnO_2_ nanocoral composite compared to those of pure NPCDs situated in 446 nm and 367 nm, respectively. However, the fluorescent intensity of the composite is dramatically decreased compared to that of pure NPCDs. This quenching occurs because of the FRET between the NPCDs and MnO_2_ in the composite, as described in Section 2.1.

The morphological structure and size of the NPCD-MnO_2_ composite was displayed in the SEM and TEM images (Figure 3). As observed in the SEM results at different magnifications, the NPCD-MnO_2_ composite has a flower-like architecture, composed of many nanosheets with sizes ranging from 250 to 500 nm. The SEM images show good agreement with the TEM images. Moreover, the TEM image (Figure 3D) shows the petals of the nanocoral composite are thin and connected to each other. A high-resolution (HR) TEM image (Figure 3E) indicates the nanocoral composite crystalline structure.

The XPS profile (Figure 4A) shows peaks at 138.17, 284.23, 399.31, 536.11, 640.69, and 656.63 eV, matching to P2p, C1s, N1s, O1s, Mn2p (Mn2p3/2 and Mn2p1/2), respectively. The weight contributions of Mn, C, N, O, and P are found to be 26.49%, 19.26%, 1.47%, 51.89%, and 0.88%, respectively. Figure 4B shows the high-resolution XPS profile of Mn2p. The peaks of the metallic states of Mn0, including Mn2p3/2 and Mn2p1/2, are placed in 642.59 and 654.25 eV, respectively. Because of the variety of functional groups, including amine, phosphate, carboxyl, and hydroxyl groups, on the surfaces of the NPCDs, the C1s peak can be separated into four peaks corresponding to C‒N/C‒P, C‒C/C=C, C‒O, and C=O bonds, which are placed in 285.14, 284.38, 286.46, and 288.68 eV, respectively (Figure 4C). Characteristic bonding is also analyzed in the high-resolution XPS profiles of P2p and N1s, which highly agree to those presented in our previous studies [13,17].

Figure 4D shows the Raman spectra of MnO_2_, NPCDs, and the NPCD-MnO_2_ nanocoral composite. Because of the Mn–O stretching vibration, there are two peaks at 662 and 735 cm^−1^ in the Raman spectrum of pristine MnO_2_. Structural defects and the E_2g_ phonon of sp^2^-bonded carbon atoms in a two-dimensional hexagonal lattice lead to the appearance of the disordered and graphitic bands located at 1290 and 1591 cm^−1^, respectively, in the spectrum of the NPCDs [22]. For the NPCD-MnO_2_ nanocoral composite, the characteristic disordered and graphitic bands of the NPCDs appear along with the two peaks of MnO_2_. These results further confirm the successful use of NPCDs as reductants to form MnO_2_, as well as the amalgamation of NPCDs and MnO_2_ in the nanocoral composite.

### 3.2. Sensing of GSH

#### 3.2.1. Sensing Mechanism

As fluorescent spectra in Figure 2, the intensity of the NPCD-MnO_2_ nanocoral composite drop markedly compared to that of the pristine NPCDs. This is the result of FRET, which happens because of the overlap between the absorption spectrum of the MnO_2_ component and the emission spectrum of the NPCD component in the nanocoral composite. The overlapped spectra are shown in Figure 5A.

However, when GSH is added to the solution of NPCD-MnO_2_ nanocoral composite, the fluorescent intensity of the composite recovers. In contrast, GSH cannot turn on the fluorescence of pristine NPCDs (Figure 5B). This is because MnO_2_ can be reduced effectively to Mn^2+^ through the oxidation of GSH to form glutathione disulfide (GSSG), according to the special reaction presented in Equation (1) [23,24]. The decomposition of MnO_2_ interrupts the FRET between the NPCDs and MnO_2_, thus “turning on” the fluorescence of the NPCDs.
2GSH + MnO_2_ + 2H^+^ → GSSG + Mn^2+^ + 2H_2_O (1)

#### 3.2.2. Optimization

It is well known that the stability and the intensity of fluorophores are strongly dependent on pH. Therefore, we investigated the pH influence on the fluorescent properties of the NPCD-MnO_2_ nanocoral composite. As presented in a previous study, NPCDs, which are one of the components in the nanocoral composite, exhibit a low fluorescent intensity in strongly acidic solutions. The intensity of the NPCDs reaches its maximum value and remains stable with pH from 6 to 10 [17]. Hence, in this study, the influence of pH values from 4 to 9 on the intensity of the NPCD-MnO_2_ nanocoral composite was investigated. The intensity of the composite grows steadily with pH from 4 to 6.5, then decreases from pH 7, reaching the lowest intensity at pH 9 (Figure 6). Based on these results, pH 6.5 was determined to be the optimal pH to maintain the high fluorescent intensity of the nanocoral composite. However, as discussed in Section 2.1, the reduction of MnO_2_ by GSH occurs under acidic conditions. Therefore, we chose a pH value of 6 to obtain better sensing performance.

To study the optimal time for GSH sensing, the fluorescent intensity of the NPCD-MnO_2_ nanocoral composite when adding 20 µM GSH was recorded every 5 min. Figure 7 indicates that the composite intensity raises steadily in the first 40 min. Thereafter, the intensity fluctuates slightly. Hence, 40 min was considered the optimal time for GSH sensing.

#### 3.2.3. GSH Sensing

When GSH is added, the fluorescence of the NPCD-MnO_2_ composite recovers (Figure 8. The fluorescent intensity of the composite increases steadily with increasing GSH concentrations belong to the range of 0–250 µM. A linear correlation is observed between the NPCD-MnO_2_ composite intensity and GSH levels of 20 to 250 µM. The corresponding equation is ***I*/*I*_o_ = 1.31061 + 0.00242C_M_**, which has a correlation coefficient (**R^2^) value of 0.9932**. To examine the productivity of the method, the sensing performance was repeated, and the data were plotted with standard deviations (SD). The limit of detection (LOD) was determined using the following equation:LOD = 3m/n,
where m is the SD value of the blank solution, and n is the slope of the calibration curve.

With values of m and n of 0.00239 and 0.00242, respectively, the LOD was calculated to be approximately 1 µM. Table 1 presents a comparison between the sensing results of our method and those of others.

#### 3.2.4. GSH Sensing in Human Serum

GSH sensing performance in human serum was measured to confirm the feasibility of the NPCD-MnO_2_ nanocoral composite-based sensor. Table 2 indicates that the recovery of GSH is 94.5%–108.4% and all the relative standard deviations (RSDs) are less than 4% with three repetitions of the experiment. This performance approves the potential of using NPCD-MnO_2_ nanocoral composite for the practical detection of GSH.

#### 3.2.5. Selective Investigation

High selectivity is a necessary requirement for sensor use. To evaluate the NPCD-MnO_2_ composite as specific to the sensing of GSH, the fluorescent responses of carbohydrates, amino acids, and ions were investigated with and without adding GSH. Figure 9 displays the addition of carbohydrates, other amino acids, and ions does not enhance the fluorescent intensity of the composite. However, when GSH co-exists with the other substances, the fluorescent intensity is recovered. This indicates that it is impossible for the other substances to interfere with the ability of GSH to “turn on” the fluorescence. The high denticity of GSH, which typically consists of two or more parts of the molecule (–SH and –COO^−^), is in charge of the “turn on” phenomenon. This chelation makes the interactions of the metal atom and the GSH stronger than that of other substances possessing either a single –SH, weak amine, or –COO^-^ binding group [33]. Therefore, the GSH– MnO_2_ interaction is much stronger than that of the others investigated. In addition, as mentioned in Section 2.1, GSH can reduce MnO_2_ to Mn^2+^, thus interrupting the FRET between the NPCDs and MnO_2_ and leading to recovery of the fluorescent intensity of the NPCDs. The considerable enhancement in the fluorescent intensity of the composite with the addition of GSH ensures that the sensing method replied on the NPCD-MnO_2_ composite is specific for GSH.

## 4. Conclusions

In conclusion, we successfully applied NPCDs as a reductant to form MnO_2_ from KMnO_4_ and the subsequent nanocoral composite. We relied on the fluorescent quenching property of the NPCD-MnO_2_ nanocoral composite to design a “turn on” fluorescence sensor for the GSH evaluation. Steady recovery in the fluorescent intensity of the NPCD-MnO_2_ nanocoral composite was observed with increasing GSH concentration. A linear correlation was obtained between the fluorescent intensity and GSH concentration from 20 to 250 µM, with an R^2^ value of 0.9932. The LOD was determined to be 1 µM. The effectiveness, selectivity, and simplicity of fabrication and operation render the NPCD-MnO_2_ nanocoral composite-based GSH sensor a potential method for GSH detection and contributes to the development of a new strategy for simple sensing approaches in biomedical application.

## Data Availability

Not applicable.

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
