# Peer review of "Glutathione Fluorescence Sensing Based on a Co-Doped Carbon Dot/Manganese Dioxide Nanocoral Composite"

_materials, 2022, doi:10.3390/ma15238677_

Round 1
Reviewer 1 Report
Dear Authors,
Concerning to your manuscript:
Glutathione fluorescence sensing based on a co-doped carbon dot/manganese dioxide nanocoral composite.
In your manuscript I found interesting and useful results about a co-doped MnO2 nanocoral composite as Glutathione fluorescence sensor which concentration range, effectiveness and selectivity of detection were determined and analyzed. These results combined whit the simplicity of fabrication and operation of the device constitute a new strategy for biomedical application. The manuscript is well supported, organized and the results were correctly analyzed and discussed. Because the results and the device describe in the manuscript are very useful I suggest its publication in MDPI Materials taking in mind only a few suggestions to improve the comprehension of all the manuscript.
1- The use of an acronym (DI) for deionized water is excessive and could be causes of confusion.
2- Figure 8 is missing.
3- Error bars are not included in the sensing graphs. Furthermore, the equation deduced from the linear region and the correlation coefficient should be included in figure 9 or highlighted in the text.
4- In Figure 10 the identity of the added substances is no clear. Should be writhe of higher size or include in the caption.
5- As summary, have the authors an structural idea of the fluorescent and the no emissive chemical species?
6- Conclusions have many experimental and results that can be optimized.
Sincerely with the best regards,
The Reviewer
Author Response
For Referee 1
Dear Referee:
We would like to thank you very much for your professional comments on our manuscript (Manuscript ID: materials-2018732). On the basis of the referees’ esteemed comments, we have made revisions to the manuscript. We hope that the revised version of the manuscript can meet the high requirements by Materials, and that you could support us. Our responses to your comments are listed below. (Answers are marked in blue).
Comment 1: The use of an acronym (DI) for deionized water is excessive and could be causes of confusion.
Reply: Thank you for your comment. To avoid an confusion for reader, we used full phrase of “deionized water” instead of DI in the revision manuscript.
Comment 2: Figure 8 is missing.
Reply: Thank you for your comment. We modified this mistake in the manuscript
Comment 3: Error bars are not included in the sensing graphs. Furthermore, the equation deduced from the linear region and the correlation coefficient should be included in figure 9 or highlighted in the text.
Reply: Thank you for your comment.
First, Figure 9 is changed to Figure 8 in the revision manuscript.
We also think that it is very necessary to repeat the sensing experiment several times to study the reproducibility of the method. Therefore, in our research, the GSH sensing experiment was repeated three times and the results along with adding error bars is presented in Figure 8.
It is different to sensing experiment, the experiments of determining optimal conditions were carried out only one time. We suppose that it is enough to determine the optimal range of pH and time. Hence, there are no error bars in the graphs presented in Figure 6 and Figure 7.
The equation deduced from the linear region and the correlation coefficient were highlighted in the text of the revision manuscript.
Comment 4: In Figure 10 the identity of the added substances is no clear. Should be written of higher size or include in the caption.
Our reply: Thank you so much for your comment. We made a modification by writing added substances in the caption as presented in the revision manuscript.
Comment 5: As summary, have the authors an structural idea of the fluorescent and the no emissive chemical species?
Our reply: NPCDs are in charge of fluorescent source. We explain the structure of NPCDs and “on-off” fluorescent mechanism in our previous research [1]
- Le, T. H.; Lee, H. J.; Kim, J. H.; Park, S. J., Detection of Ferric Ions and Catecholamine Neurotransmitters via Highly Fluorescent Heteroatom Co-Doped Carbon Dots. Sensors 2020, 20, (12).
Comment 6: Conclusions have many experimental and results that can be optimized.
Our reply: Thank you so much for your valuable suggestion. We are going to continue to develop carbon dots-based composite material as well as optimize conditions to obtain more effective results for GSH sensing and other sensing in the further research.

Reviewer 2 Report
The sensing property of NPCD-MnO2 as a GHS sensor was reported in this manuscript. But I don’t think it is publishable because of the following reasons:
1. There are no experimental results to support the sensing mechanism.
2. The recovery of the fluorescence intensity in the presence of GHS is too low compared to that of NPCD, which means that the sensitivity and mechanism are all doubtable.
3. Does figure 9B show the relationship between the fluorescent intensity of NPCD-MnO2 and the concentration of GHS?
Author Response
For Referee 2
Dear Referee:
We would like to thank you very much for your professional comments on our manuscript (Manuscript ID: materials-2018732). On the basis of the referees’ esteemed comments, we have made revisions to the manuscript. We hope that the revised version of the manuscript can meet the high requirements by Materials, and can make you reconsider your decision. Our responses to your comments are listed below. (Answers are marked in blue).
Comment 1: There are no experimental results to support the sensing mechanism.
Reply: The GSH sensor was fabricated based on “off-on” fluorescent mechanism of NPCD-MnO2 nanocoral composite. The overlap between the absorption spectrum of MnO2 component and the fluorescence emission spectrum of the NPCD component in the composite leads to effective fluorescence resonance energy transfer (FRET), causing a decrease in the fluorescent intensity of the NPCDs. When GSH is added to the solution, an redox reaction is happened between GSH and MnO2, thus GSH can interrupt the FRET of NPCDs and MnO2 and then “turn on” the fluorescence of NPCDs. This mechanism is supported by the results presented in Figure 2, Figure 5, and Figure 8.
Comment 2: The recovery of the fluorescence intensity in the presence of GHS is too low compared to that of NPCD, which means that the sensitivity and mechanism are all doubtable.
Reply: As shown in Figure 5, there is no signal of fluorescent response when GSH is added to the pristine NPCD solution. This indicates that the pristine NPCDs can not be used to detect GSH though they have strong fluorescence. Therefore, although the fluorescent recovery of NPCD-MnO2 nanocoral composite in the presence GSH is too low compared to that of the pristine NPCDs, the NPCD-MnO2 is used to detect GSH. The fluorescent “turn on” signal is very clear in case of adding GSH to the NPCD-MnO2 solution. And when we consider about the fluorescence “turn on” ability of GSH, we only compare the intensity of NPCD-MnO2 in the presence and absence of GSH. In the other word, it is not related to the fluorescence of the pristine NPCDs.
Comment 3: Does figure 9B show the relationship between the fluorescent intensity of NPCD-MnO2 and the concentration of GHS?
Reply: After make a modification in the revision manuscript, Figure 9 is changed to Figure 8.
Figure 8B shows the relative fluorescent intensity of the NPCD-MnO2 composite in the presence of different levels of GSH. With the GSH concentration ranges from 0 to 250 µM, the fluorescence of the NPCD-MnO2 increases steadily, but there is no clear relationship. However, with the concentration from 20 to 250 µM, a linear relationship between the fluorescent intensity of the NPCD-MnO2 and the concentration of GSH is obtained with the equation is I/I0 = 1.31061 + 0.00242CM and (R2) value of 0.9932. This linear relationship is shown in inset of Figure 8B.

Reviewer 3 Report
Dear Authors.
Thank you for such an interesting article, very well approached from the experimental point of view, with conclusive results, but that can be improved with some bibliographic data, and answering some doubts that I had after reading your manuscript.
The Introduction section is well done and to allowed understand the context. Also, it would be necessary for this section or discussion, to see if the literal range of detection by NPCD-MnO2 is within physiological or pathological ranges in humans. This is where this novel methodology is finally to be applied.
The methodology is very well achieved, and it is possible to understand the particularities and how they were carried out. However, there is no statistical analysis section, and throughout the results curve fits are shown, with a calculation of R, but there is no statement of what type of fit was achieved, at least not in the methodology section.
Finally, they test it in human serum, but the conditions of that fluid, healthy patients, with some prevalent disease, are not stated in the manuscript. It would give more value to your work if you would test it in other fluids, plasma, tissues, etc.
I approve your publication, with minor observations, such as the importance of comparing your detection levels and linear range, with GSH concentrations published by other authors, with other methodologies.
Author Response
For Referee 3
Dear Referee:
We would like to thank you very much for your professional comments on our manuscript (Manuscript ID: materials-2018732). On the basis of the referees’ esteemed comments, we have made revisions to the manuscript. We hope that the revised version of the manuscript can meet the high requirements by Materials, and that you could support us. Our responses to your comments are listed below. (Answers are marked in blue).
Comment 1: The Introduction section is well done and to allowed understand the context. Also, it would be necessary for this section or discussion, to see if the literal range of detection by NPCD-MnO2 is within physiological or pathological ranges in humans. This is where this novel methodology is finally to be applied.
Reply: Thank you for your valuable comment. As your suggestion, we add some information of GSH concentration in the healthy human body in the revision manuscript. The defficiency of GSH relates to variety of diseases and we don’t have the exact information of GSH level of patients of these diseases, yet. However, according to the information of GSH concentration in the healthy human body as well as compared to the range of GSH concentration range investigated in the previous research, we think that the GSH range we carried out in our research is suitable.
Comment 2: The methodology is very well achieved, and it is possible to understand the particularities and how they were carried out. However, there is no statistical analysis section, and throughout the results curve fits are shown, with a calculation of R, but there is no statement of what type of fit was achieved, at least not in the methodology section.
Reply: Thank you for your comment. For sensing experiment, the experiment for each concentration is repeated three times (which is mentioned in the method section) to calculate the standard deviation (SD) and relative standard deviation (RSD) values. The SD values are shown via error bars in Figure 8B. Besides, with the statistical analysis we obtained, all RSD values are smaller than 3%. This indicates that all results are acceptable.
In the manuscript, we stated that a linear relationship between the fluorescent intensity of the NPCD-MnO2 composite and the concentration of GSH is obtained when the level of GSH ranges from 20 to 250 µM, with the equation is I/I0 = 1.31061 + 0.00242CM and (R2) value of 0.9932.
Comment 3: Finally, they test it in human serum, but the conditions of that fluid, healthy patients, with some prevalent disease, are not stated in the manuscript. It would give more value to your work if you would test it in other fluids, plasma, tissues, etc.
Reply: Thank you for your comment. Human serum was obtained from Sigma Aldrich with the information shown in the word attached file. It indicates that the used human serum belongs to an healthy man.
We totally agree with you that it will be more valuable if we test in other fluids, plasma, or tissues. However, in the present, we don’t have enough condition to carry out sensing experiment in plasma or tissues. We hope that we can do these experiment in the further sensing research.

Round 2
Reviewer 2 Report
I suggest accepting this manuscript. However, there are still several questions the authors should correct.
1. There are no direct experimental results to support the sensing mechanism. The overlap between the absorption spectrum of MnO2 component and the fluorescence emission spectrum of the NPCD component is just an indirect evidence. If the redox reaction is happened between GSH and MnO2, the increase of the concentration of Mn2+ will increase, and the product of GSSG should be detectable, the authors should add some experimental data to support the raised mechanism.
2. The X-axes in figure 8B should be the concentration of GHS. However, it is still labelled as wavelength.
3. The recovery of the fluorescence intensity in the presence of GHS is low compared to that of NPCD, which means that the GHS only can decompose a small amount of MnO2. Why most of the MnO2 is not reduced? Or there are some other reasons.
Author Response
Comment 1: There are no direct experimental results to support the sensing mechanism. The overlap between the absorption spectrum of MnO2 component and the fluorescence emission spectrum of the NPCD component is just an indirect evidence. If the redox reaction is happened between GSH and MnO2, the increase of the concentration of Mn2+ will increase, and the product of GSSG should be detectable, the authors should add some experimental data to support the raised mechanism.
Reply: Thank you so much for your comment. Actually, your recommend is very valuable for our research. We totally agree with you that if we add some experiment data to indicate the increasing of Mn2+ and GSSG, the mechanism will be explained clearer. However, we only have short time for the second round of revision so we are so sorry that this time we can not add more data to support the mechanism.
We are going to continue to develop carbon dot-based composite material as well as effective fluorescence methods to detect GSH and other substances for further research. That time we are going to carry out the kind of experiments as you mentioned to support mechanism explanation.
Comment 2: The X-axes in figure 8B should be the concentration of GSH. However, it is still labelled as wavelength.
Reply: Thank you so much for your careful checking. I modified this mistake in the revised manuscript.
Comment 3: The recovery of the fluorescence intensity in the presence of GSH is low compared to that of NPCD, which means that the GSH only can decompose a small amount of MnO2. Why most of the MnO2 is not reduced? Or there are some other reasons.
Reply: Thank you so much for your comment.
When we use the strong reductants such as strong metal, strong acids, MnO2 can not be totally reduced. There is always a remain amount of MnO2 is not reduced.
NPCDs can be used as a reductant, NPCDs can decompose a part of the MnO2 amount. However, compare to the other reductants, NPCDs are friendly to environmental as well as biological application. Moreover, there is no excess ion of reductants existing in the composite material. Hence, using NPCDs as a reductant is acceptable.
